# Sex Differences in Complex Posttraumatic Stress Disorder Network among Chinese Young Adults

**DOI:** 10.3390/bs13100846

**Published:** 2023-10-16

**Authors:** Yiming Liang, Luxi Yang

**Affiliations:** Shanghai Key Laboratory of Mental Health and Psychological Crisis Intervention, Affiliated Mental Health Center (ECNU), School of Psychology and Cognitive Science, East China Normal University, Shanghai 200062, China

**Keywords:** complex posttraumatic stress disorder, network analysis, sex differences, China

## Abstract

Evidence suggests that sex differences commonly occur in trauma-related disorders. The current study aims to explore sex differences in complex posttraumatic stress disorder (CPTSD) symptom networks among Chinese young adults with childhood trauma. The current study utilized a representative sample of college students in Beijing and included 1416 participants (409 men and 907 women) who had childhood trauma experience. CPTSD symptoms were evaluated using the International Trauma Questionnaire. Regularized partial correlation network analysis and Bayesian network analysis were used to estimate the network structure and possible causality of CPTSD symptoms for both sexes. Male and female CPTSD symptom networks had differences in strength centrality and bridge centrality. Nightmares and feelings of failure had the highest strength centrality, and long-term upset and nightmares had the highest bridge centrality for men. Hypervigilance and feelings of failure had the highest strength centrality, and long-term upset and exaggerated startle response had the highest bridge centrality for women. The current study provides the first evidence of sex differences in the CPTSD symptom network among Chinese young adults with childhood trauma. Young men and women differed in highly central symptoms, which may speak to sex specificity in the main manifestations of CPTSD symptoms.

## 1. Introduction

Complex posttraumatic stress disorder (CPTSD) was first proposed by Lewis Herman in her study of women victims who had experienced chronic domestic and sexual violence [1]. Under discussion over the years, the World Health Organization introduced CPTSD as an independent diagnosis in the 11th Revision of International Classification of Diseases (ICD-11) [2]. In the diagnostic criteria of trauma-related mental disorders, the ICD-11 introduced CPTSD as a sibling disorder to posttraumatic stress disorder (PTSD). Individuals with CPTSD not only have the core symptoms of PTSD (reexperience, avoidance and sense of threat) but also suffer from “disturbances in self-organization” (DSO: affective dysregulation, negative self-concept and difficulties in relationships). These DSO symptoms characterize the impact of trauma on individuals’ emotion regulation, identity and interpersonal relationships [3]. Considering that the proposal of ICD-11 CPTSD is relatively recent, the empirical testing of its symptom structure is of great importance.

Unlike the fifth edition of the Diagnostic and Statistical Manual of Mental Disorders (DSM-5), the type of trauma is not a prerequisite for being diagnosed with CPTSD or PTSD in diagnostic criteria of ICD-11 [4]. Nevertheless, previous studies found that CPTSD is more likely associated with prolonged and repeated trauma types, including those which commonly happened in childhood, such as physical abuse or sexual trauma [5,6,7]. A study further suggests that adverse childhood experiences remain uniquely predictive of CPTSD symptoms after controlling for lifelong traumatic stress. Some studies have also shown that some single life-changing accidents can cause CPTSD, because these trauma types may lead to violations of the integrity of the body and, in turn, victims are unable to reexperience positive identities [3,8].

Borsboom [9] proposed a new perspective on the conceptualization of mental disorders, that is, psychopathological network theory. This theory defines mental disorders as a series of interacting symptoms [10]. The triggering of one symptom may lead to the activation of other symptoms. In this way, the dynamic causality between symptoms constitutes the essence of mental disorders. The most common model used to estimate psychological networks is the partial correlation network [11]. Network graph visualization makes the links among symptoms comprehensible at a glance and clarifies these links. In addition, identifying central and bridge symptoms is also valuable. Centrality, as a property index of each node, gives details about which symptoms are the most interconnected with the other symptoms in the network. The arousal of a central symptom might drive the continued activation of other symptoms, highlighting its critical diagnostic implications [12]. Bridge symptoms connect different disorders or different subgroups within the same disorder [13]. How the two symptom clusters of CPTSD are related is also a crucial issue that can be examined from the bridge symptom perspective.

However, only six studies have explored symptom networks in CPTSD using the network approach so far [14,15,16,17,18,19]. Knefel and colleagues found similarities among CPTSD networks in adult samples from Germany, Israel, the UK and the USA [15]. This study showed that negative self-concept was the central symptom, especially for individuals with childhood trauma experiences [9]. Knefel and colleagues further examined CPTSD networks in adult samples of Austria, the UK and Lithuania, and found that a feeling of worthlessness, one form of negative self-concept, was the most central symptom [16]. This remarkable finding in terms of central symptoms was proven repeatedly by Levin and colleagues [17]. Nevertheless, there is a lack of investigation into the CPTSD network in Asian cultures. More importantly, sex differences in CPTSD have not been studied.

The question of sex differences in CPTSD is of theoretical and practical significance, but sex difference in the network structure of symptoms had not been assessed. Traditionally, researchers have paid particular attention to sex differences in mental disorders [20,21]. Sex differences in trauma-related disorders are multifactorial; differences in the type of trauma that men and women are likely to be exposed to and their reactivity after trauma are important social psychological factors [22,23]. Sex differences in the symptomatology of PTSD have been documented in previous epidemiological and clinical studies. Women seem to be more sensitive to the disorder, more likely to meet the diagnostic criteria [24] and more likely to experience a chronic course of the disease [25,26,27]. Another significant sex difference in PTSD exists in the manifestation of PTSD symptoms. For example, women score higher on numbness, avoidance, re-experiencing and hyperarousal [28], while men report nightmares and irritability more frequently [29]. Due to the recent proposal of the diagnostic criteria for CPTSD, there is little literature on sex differences in CPTSD. The results of the existing literature on sex differences in CPTSD are mixed [5], with most of studies indicating that females have a greater risk of CPTSD than males [30,31].

To increase our understanding of sex differences in the CPTSD network, the current study takes the lead in providing evidence for such differences in the pattern of CPTSD symptoms through network analysis. We intend to explore the potential sex differences in network structure and key symptoms in CPTSD.

## 2. Methods

### 2.1. Participants and Procedure

The participants recruited in this study were college students in Beijing, China. To obtain a representative sample of Beijing college students, we employed a random stratified sampling procedure. Taking the discipline types of universities as the classification standard, these universities were divided into 13 types. We also took the different running levels of universities into account at the same time, trying to cover both key universities and ordinary universities under various types of universities as much as possible. Finally, the distribution of the 31 universities in this study was as follows: comprehensive (5), Science (5), Engineering (5), Agriculture (2), Normal (2), Finance and Economics (3), Forestry (1), Politics and Law (1), Medicine (1), Language (3), Nationality (1), Art (1) and Sports (1). The students were divided into strata in advance based on their universities, majors (liberal arts or sciences) and grades. Moreover, in the selected universities, we further stratified sampling by grade to ensure the diversity and representativeness of participants.

Then, we contacted some teachers in the colleges that were our sampling targets and asked for their assistance in distributing questionnaires until the number of answer sheets in each stratum reached the recruitment number. Each participant signed an informed consent form before accessing and completing our questionnaires online. If any of the participants felt uncomfortable after filling out questionnaires, our research team provided psychological services to them through a psychological hotline. The present study was conducted in accordance with the Declaration of Helsinki and approved by the Ethics Committee for Human Research of East China Normal University.

A total of 2048 participants from 29 universities completed the survey. Among them, 221 participants were excluded due to careless answering (e.g., answering the same answer to each item or failure to pass the attention check items). After that, 1827 (89.2%) valid data were further screened: 411 participants who did not report childhood trauma history were further excluded according to the Life Events Checklist for DSM-5. Finally, 1416 met the inclusion criteria (i.e., having childhood trauma and being aged 18–28 years). The participants’ mean age was 20.60 years (*SD* = 1.40), and 64.1% of them were women. The prevalence of childhood trauma is shown in Table 1. The three most common trauma types were physical assault, transportation accidents and natural disasters.

### 2.2. Measurements

#### 2.2.1. CPTSD

The International Trauma Questionnaire (ITQ) was a reliable self-report measure of ICD-11 PTSD and CPTSD symptoms [32]. The Chinese version of the ITQ was translated by Ho and colleagues and has been demonstrated to have good reliability and validity (Cronbach’s α of the six PTSD items = 0.89; Cronbach’s α of the six DSO items = 0.90) [33]. Six items assess PTSD symptoms, including reexperience (RE), avoidance (AV) and sense of threat (TH), with each symptom measured by two items, and three items are used to assess functional impairment associated with PTSD symptoms. Similarly, six items assess DSO symptoms, including affective dysregulation (AD), negative self-concept (NSC), and difficulties in relationship (DR), with each symptom measured by two items; functional impairment caused by DSO symptoms is assessed with three items. All items in the ITQ are scored on a 5-point scale (0 = not at all, 4 = extremely), and scores ≥2 indicate the presence of a symptom or functional impairment. In this study, the Cronbach’s α was 0.92 for men and 0.90 for women.

#### 2.2.2. Childhood Trauma

The Life Events Checklist for DSM-5 (LEC-5) was used to assess childhood trauma. The original version of the LEC-5 consists of 17 traumatic events [34]. In the current study, we removed 4 events that are very unlikely to happen to Chinese college students (i.e., experiencing war, being imprisoned). Each item was scored on a six-point Likert scale, ranging from 0 (never happened to me) to 5 (extremely), and higher total scores indicated a more serious childhood trauma.

### 2.3. Data Analysis

Descriptive analysis was conducted in IBM SPSS Statistics 23.0. The prevalence of each reported traumatic event was calculated. Based on the cut-off of ITQ (scores ≥2 indicate the presence of a symptom), we reported prevalence of each CPTSD symptom. Means and standard deviations of CPTSD symptoms were also examined.

#### 2.3.1. Network Estimation

We divided the participants into men and women subsamples and followed the statistical procedure described by Epskamp and Fried [35] to estimate networks. All analyses were performed using R version 4.1.2 (R Core Team, 2013). The networks were visualized with the qgraph package [36]. A total of 0.46% of the CPTSD item-level data were missing. Missing data were handled using the maximum likelihood method for network analysis.

The symptom structure of CPTSD was estimated utilizing a partial correlation network, and the association parameters between all nodes were calculated according to Gaussian graphical models (GGMs). In GGMs, if two symptoms are connected in the outcome graph, they are related to each other after controlling for all other symptoms (i.e., partially correlated). Next, we used the least absolute shrinkage and selection operator (LASSO) regression model to set small correlations to zero [37]. This process employed a regularization rule that reports edges cautiously to ensure the simplicity and accuracy of the network structure [38]. For more details on these computational techniques, see the corresponding tutorial [35].

#### 2.3.2. Centrality Estimation

Centrality is a vital index for understanding networks. It reveals the importance of any given node in a network, that is, which symptoms can broadly activate other symptoms or are easily activated by other symptoms. Strength is the most widely used and robust centrality measure; it refers to the weighted sum of all edges connected to a particular node [39]. To determine the relationship between PTSD and DSO, the two symptom groups of CPTSD, we employed bridge strength, a network statistic developed by Jones and colleagues [40], to identify bridge symptoms.

#### 2.3.3. Network Stability

The estimation of network stability relies on the emerging method proposed by Epskamp and Fried [35]. The accuracy of edges was estimated using the R bootnet package [41] to bootstrap the 95% confidence intervals. The number of bootstrap samples was 1000. The stability of centrality indices was estimated using a subset bootstrapping procedure. The program extracted subsets from the original data, calculated node centrality based on the subsets, and reported the correlation of the centrality between the subsets and the entire sample. The correlation stability coefficients (CS coefficients) of the centrality indices were also estimated. The value ranges from 0 to 1. When it is above 0.25/0.50, it indicates that the estimation of centrality has moderate stability/strong stability [42]. Finally, we executed difference tests on both edge weights and centrality indices.

#### 2.3.4. Network Comparison

The R NetworkComparisonTest package was used to examine the potential differences between networks among men and women. We compared the global connectivity of the two networks to assess how densely the symptoms connected with each other. Furthermore, edge weights and node centrality in male and female networks were compared pairwise.

#### 2.3.5. Bayesian Network Estimation

The Bayesian network attempts to predict the direction of edges in a network. A Bayesian network is a probabilistic graphical model visualized in the form of a directed acyclic graph (DAG). Nodes in a Bayesian network are connected by directed edges (i.e., arrows), allowing causal interpretations of relationships between nodes [43]. Variables are placed in a putative causal cascade, where upstream variables are seen as the causes of downstream variables [44]. The direction of an arrow in the Bayesian network predicts how activation of one symptom will spread to another symptom. There is no feedback loop.

Estimation of the Bayesian network was performed using the hill-climbing algorithm [45] in the R package bnlearn. A bootstrapping function in bnlearn was used to construct the network structure by adding, subtracting, and reversing the edges, randomly restarting the process with a different candidate edge. As this iterative process unfolded, the algorithm gradually optimized the Bayesian information criterion (BIC) to determine the best-fit network. To ensure the stability of the Bayesian network, multiple bootstrapping samples were drawn, and the averaged network was presented as the final result [46].

## 3. Results

Among 1416 participants with childhood trauma, the prevalence rates of probable CPTSD and probable PTSD were 9.96% (n = 141) and 5.86% (n = 83), respectively. In addition, 9.60% (n = 136) of participants suffered from DSO symptoms.

Table 2 shows the mean scores, standard deviations, and prevalence of the samples for each symptom of the ITQ. Long-term upset (AD1) was the most-reported symptom, whereas feeling distant or cut off from others (DR1) were the least-reported symptom.

### 3.1. Networks and Centrality Estimation

The CPTSD symptom networks for men and women are shown in Figure 1. The associations within each symptom cluster were obviously strong, such as the associations between the two re-experiencing symptoms (RE1 and RE2) and two negative self-concept symptoms (NSC1 and NSC2). The network density of the male sample was 62.1% (41 nonzero edges out of 66 possible edges), higher than the female sample, which had a network density of 53.0% (35 nonzero edges).

Figure 2 presents the results of strength centrality analyses. The bridge strengths are summarized in Figure 3. For men, the symptoms with the strongest strength centrality were nightmares (RE1) and feelings of failure (NSC1). The strongest bridge estimates were long-term upset (AD1) and nightmares (RE1). As for the network of CPTSD among women, hypervigilance (TH1) and feelings of failure (NSC1) showed the strongest strength centralities of all symptoms. The strongest bridge symptoms were long-term upset (AD1) and exaggerated startle response (TH2). Overall, there were differences in strength centrality and bridge strength in the male and female CPTSD networks.

### 3.2. Network Accuracy and Stability

The results of edge weight bootstrapping revealed that CPTSD networks in both males and females were moderately accurately estimated (see Appendix A). In both male and female networks, although there was considerable overlap among the 95% CIs of edge weights, nonoverlapping CIs also existed. Moreover, most of the strongest edges were significantly different from most other edges in CPTSD networks (see Appendix A).

The subset bootstrap showed stable and interpretable estimates of the order of node strength centrality in the networks in both male and female samples (see Appendix A). The CS coefficients for strength and bridge strength were 0.52 and 0.44 in the male network and 0.75 and 0.59 in the female network, respectively. The results of subset bootstrapping for node strength centrality are shown in Appendix A. Moreover, centrality difference tests further favored the order of node centrality in both male and female networks. The most central nodes were statistically stronger than most other nodes in the networks (see Appendix A).

### 3.3. Network Comparison

The results of the network comparison test (NCT) indicated global strengths of 5.57 and 5.40 for men and women, respectively, and the difference in global connectivity of the two networks reached statistical significance (*S* = 0.17, *p* = 0.004). NCT also showed that 21.6% of the edges in the networks were significantly different in the two networks. Moreover, NCT also showed that RE1 (upsetting dreams) and long-term upset (AD1) had significantly different strength centralities in the two samples.

### 3.4. Bayesian Network

The Bayesian network was used to predict the probable direction of edges in men and women networks. Upstream variables are seen as the causes of downstream variables in Bayesian networks, so upstream symptoms have a greater likelihood of activating the whole network. The results of the Bayesian network suggested upsetting dreams (RE1) as a key driver of other symptoms (see Figure 4a) in the men sample and hypervigilance (TH1) as the only key symptom directly or indirectly driving most of other symptoms in women (see Figure 4b). Long-term upset (AD1) was a bridge between PTSD symptoms and DSO symptoms in both men and women samples.

## 4. Discussion

To our knowledge, the present study provides the first evidence of sex differences in the CPTSD symptom network. The findings showed that male and female CPTSD symptom networks had differences in strength centrality and bridge centrality. Nightmares and feelings of failure had the highest strength centrality, and long-term upset and nightmares had the highest bridge centrality for men. Hypervigilance and feelings of failure had the highest strength centrality, and long-term upset and exaggerated startle response had the highest bridge centrality for women.

Both men and women have higher strength centrality in negative self-concept symptoms of DSO, and a feeling of failure was among the more central nodes both in men and women the CPTSD network. Previous research has also found that negative self-concept of DSO was the central symptom in the CPTSD network [15,16], which have previously been considered central to the diagnosis of CPTSD [31]. Negative self-evaluations, such as self-deprecation and self-loathing, were observed to be associated with a higher risk in individuals with a childhood trauma experience [3]. These people scored lower on emotional self-esteem as well [47]. However, the results of the Bayesian network showed that negative self-concept symptoms were downstream of CPTSD symptoms in both male and female networks, which indicated that negative self-concept symptoms may be activated in CPTSD networks. Several studies have demonstrated that long-term interpersonal trauma, especially in childhood, has a strong negative impact on the development of self-esteem [48]. This evidence supports that a negative self-concept may be an “outcome” symptom in CPTSD.

Among the male group, nightmares had the highest strength centrality and a high level of bridge strength. The results of the Bayesian network also indicated that nightmares were a key driver in male CPTSD symptoms. Previous studies found that nightmares had high strength centrality in the PTSD symptom network among male-dominated samples [49], such as military samples [50]. Taken together, these findings perhaps imply that symptoms of nightmares have a male-specific role in trauma-related disorders. In contrast, hypervigilance had the highest strength centrality and was a key driver in the Bayesian network among the female sample, which indicated that hypervigilance might a more important posttraumatic stress manifestation in women. Previous research on PTSD symptom development patterns has shown that women typically score higher than men on subjective arousal during the acute phase of stress disorder development, and threat perception is an important aspect [49]. Women tend to use more emotion-focused, defensive, and palliative coping styles than men in stressful situations [51]. This may explain why hypervigilance plays a greater role in the CPTSD network among women.

In terms of bridge strength, long-term upset was found to be a crucial bridge symptom for both men and women. PTSD is highly associated with mood disorders [52], while DSO mainly refers to some functional dysregulation, including emotional dysregulation. Disturbance in emotional function might form connections between the two parts of CPTSD. Individuals with traumatic experiences show distressed emotions such as sadness and frustration, which are also deemed to be the emotional states existing in life with long-term negative self-concept and poor relationships [53,54]. In addition to the common ground on sex, nightmares were identified as another bridge symptom in the male network, indicating its unique role to men.

The present study had several limitations. First, the current study only focused on young Chinese adults. However, the characteristics of the CPTSD network in adults of other age groups need to be investigated in the future. Second, although ITQ is proven to be an effective tool for measuring CPTSD [26], the natural restrictions of self-report questionnaires still exist. The structured interviews by clinicians remain to be used to provide more valid criteria for investigating symptoms. Third, although this study used a Bayesian network for the causal prediction of symptoms, the temporal order of the CPTSD symptoms is still unknown. Future studies should employ longitudinal network analysis methods to clarify the causal relationships of CPTSD symptoms. Finally, the composition of the sample in our study has a gender skew, with approximately two-thirds of the participants being women. Caution is advised when inferring the results of NCT.

Despite these limitations, this study provides the first insight into sex differences in the CPTSD symptom network among Chinese young adults with childhood trauma. The finding that young men and women differed in highly central symptoms may speak to sex specificity in the main manifestations of CPTSD symptoms. For the diagnosis of CPTSD in clinical work, attention should also be paid to different symptoms according to sex. For males, more attention should be paid to intrusive symptoms such as nightmares, while more attention should be paid to hypervigilance symptoms for females.

## 5. Conclusions

The current study found that the sex differences in the CPTSD networks were mainly manifested in the strength centrality and bridge strength of symptoms. Nightmares and feelings of failure had the highest strength centrality for men, while hypervigilance and feelings of failure had the highest strength centrality for women. Long-term upset and nightmares had the highest bridge centrality for men, while long-term upset and exaggerated startle response had the highest bridge centrality for women. According to these findings, attention should also be paid to different CPTSD symptoms according to sex in clinical work.

## Figures and Tables

**Figure 1 behavsci-13-00846-f001:**
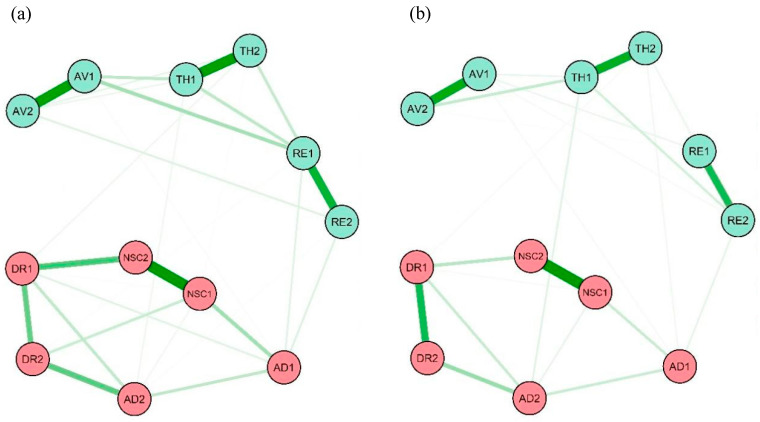
CPTSD symptom networks for men (**a**) and women (**b**). Notes: RE1: nightmares; RE2: flashbacks; AV1: internal avoidance; AV2: external avoidance; TH1: hypervigilance; TH2: exaggerated startle response; AD1: long-term upset; AD2: emotional numbing; NSC1: feelings of failure; NSC2: feelings of worthlessness; DR1: feeling distant or cut off from others; DR2: difficulties feeling close to others.

**Figure 2 behavsci-13-00846-f002:**
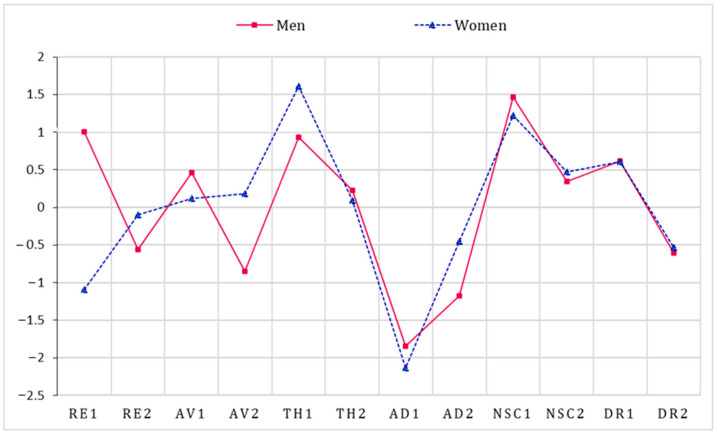
Standardized strength centrality in men and women samples. Notes: RE1: nightmares; RE2: flashbacks; AV1: internal avoidance; AV2: external avoidance; TH1: hypervigilance; TH2: exaggerated startle response; AD1: long-term upset; AD2: emotional numbing; NSC1: feelings of failure; NSC2: feelings of worthlessness; DR1: feeling distant or cut off from others; DR2: difficulties feeling close to others.

**Figure 3 behavsci-13-00846-f003:**
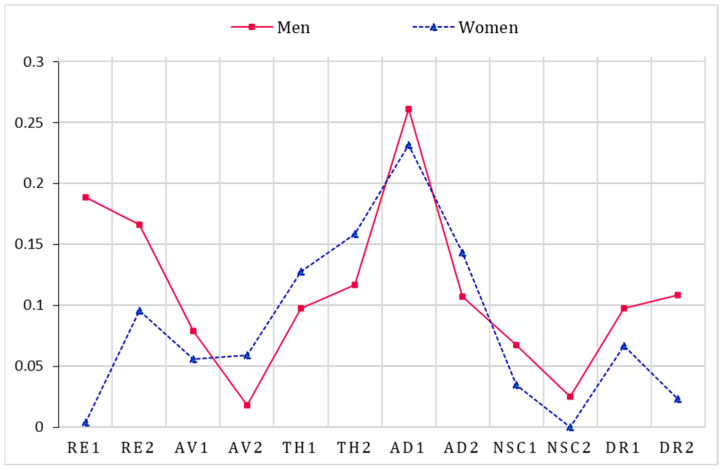
Bridge strength of symptoms within men and women samples. Notes: RE1: nightmares; RE2: flashbacks; AV1: internal avoidance; AV2: external avoidance; TH1: hypervigilance; TH2: exaggerated startle response; AD1: long-term upset; AD2: emotional numbing; NSC1: feelings of failure; NSC2: feelings of worthlessness; DR1: feeling distant or cut off from others; DR2: difficulties feeling close to others.

**Figure 4 behavsci-13-00846-f004:**
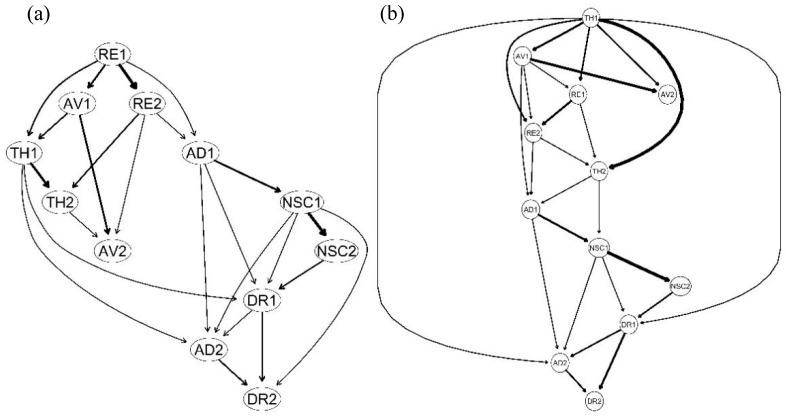
Directed acyclic graphs of men (**a**) and women (**b**). Notes: RE1: nightmares; RE2: flashbacks; AV1: internal avoidance; AV2: external avoidance; TH1: hypervigilance; TH2: exaggerated startle response; AD1: long-term upset; AD2: emotional numbing; NSC1: feelings of failure; NSC2: feelings of worthlessness; DR1: feeling distant or cut off from others; DR2: difficulties feeling close to others.

**Table 1 behavsci-13-00846-t001:** Prevalence of the reported childhood traumatic events.

Event	Respondents	Percentage of the Sample
Natural disasters	481	33.97%
Transportation accident	572	40.40%
Physical assault	605	42.73%
Life-threatening illness or injury	343	24.22%
Unwanted or uncomfortable sexual experience	326	23.02%
Causing serious injury or death to someone else	314	22.18%
Assault with a weapon	266	18.79%
Sudden accidental death	156	11.02%
Fire or explosion	268	18.93%
Sudden death to a loved one	309	21.82%
Exposure to toxic substance	100	7.06%
Sexual assault	73	5.16%

**Table 2 behavsci-13-00846-t002:** Mean scores, standard deviations and prevalence of CPTSD symptoms.

Symptom	*M*	*SD*	Percentage
RE1	1.00	1.10	28.46%
RE2	1.14	1.19	36.51%
AV1	1.48	1.27	44.07%
AV2	1.41	1.31	41.81%
TH1	1.02	1.24	30.16%
TH2	1.05	1.21	30.37%
AD1	1.60	1.06	47.46%
AD2	1.25	1.22	37.50%
NSC1	1.20	1.25	35.88%
NSC2	0.97	1.21	28.46%
DR1	0.98	1.15	27.40%
DR2	1.42	1.30	36.51%

Notes: RE1: nightmares; RE2: flashbacks; AV1: internal avoidance; AV2: external avoidance; TH1: hypervigilance; TH2: exaggerated startle response; AD1: long-term upset; AD2: emotional numbing; NSC1: feelings of failure; NSC2: feelings of worthlessness; DR1: feeling distant or cut off from others; DR2: difficulties feeling close to others.

## Data Availability

The data presented in this study are available on request from the corresponding author.

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
