# Peer review of "Sex Differences in Complex Posttraumatic Stress Disorder Network among Chinese Young Adults"

_behavsci, 2023, doi:10.3390/bs13100846_

Round 1

Reviewer 1 Report

Title: Sex Differences in Complex Posttraumatic Stress Disorder Networks Among Chinese Young Adults

Thank you for providing me with this article for review. I believe that addressing Complex Posttraumatic Stress Disorder (CPTSD) is a crucial step in our understanding of this syndrome, and it warrants special attention, possibly requiring specific treatment approaches tailored to this subgroup within the broader category of PTSD.

In this case, the article at hand is a significant contribution to the field. While it doesn't delve into practical treatment strategies, some of its findings can serve as valuable guidance for therapists dealing with CPTSD.

I have several concerns that I would like to highlight:

  1. The composition of the sample in this study exhibits a noticeable gender skew, with approximately two-thirds of the participants being female.
  2. To my understanding, CPTSD is fundamentally characterized by its protracted duration. However, the absence of any reference to this temporal dimension in the study is noteworthy. The utilization of the Life Events Checklist for DSM-5 (LEC-5) in the research is acknowledged; however, its intended function, as delineated in its manual, is primarily the collection of data concerning potentially traumatic experiences. It does not proffer a formal scoring protocol or interpretation thereof. Hence, it remains unclear how the researchers identified and differentiated cases of CPTSD without factoring in the prolonged or repetitive nature of the traumatic experiences.
  3. According to the ICD-11, complex PTSD typically results from exposure to traumatic events of an extremely threatening nature, often characterized by their prolonged or repetitive nature and the impossibility of escape. I would appreciate more information on how the researchers established the presence of CPTSD in their sample, especially considering that the most frequent incidents reported were natural disasters, which generally have a lower probability of resulting in PTSD; transportation accidents, which are typically single events unless prolonged hospitalization or when injury or death of next of kin occurred; and physical assaults, which in many cases are isolated events.
  4. Notwithstanding these reservations, I find this article to be intellectually stimulating, with the statistical methodologies employed therein both enlightening and innovative. Consequently, I anticipate the authors' forthcoming response addressing the concerns articulated above.
  5. Furthermore, I posit that the findings pertaining to gender-specific manifestations of CPTSD symptoms, notably the prevalence of nightmares and feelings of failure among men and the prevalence of hypervigilance, long-term distress, and exaggerated startle responses among women, carry immediate clinical relevance. This understanding could empower clinicians to deploy targeted interventions aimed at alleviating these symptoms, potentially enhancing the effectiveness of CPTSD treatments.
  6. Overall, I appreciate the article's contribution to the field and eagerly await the authors' response to these comments.

Author Response

Dear Reviewer 1,

Thank you very much for giving us the opportunity to revise our manuscript entitled “Sex differences in complex posttraumatic stress disorder network among Chinese young adults” (Manuscript Number: behavsci-2613111). We greatly appreciate your comments and suggestions. We have revised the manuscript and would like to resubmit it for publication in the Behavioral Sciences. We hope that the revised manuscript addresses most of the concerns that you raised and incorporates most of your suggestions. We hope to receive a favorable response from you. Please find our point-by-point responses attached.

Responses to Reviewer 1

Thank you for providing me with this article for review. I believe that addressing Complex Posttraumatic Stress Disorder (CPTSD) is a crucial step in our understanding of this syndrome, and it warrants special attention, possibly requiring specific treatment approaches tailored to this subgroup within the broader category of PTSD.

In this case, the article at hand is a significant contribution to the field. While it doesn't delve into practical treatment strategies, some of its findings can serve as valuable guidance for therapists dealing with CPTSD.

I have several concerns that I would like to highlight:

Response: Thank you for your careful reading and encouragement. We appreciate your comments and have revised the manuscript accordingly.

1.The composition of the sample in this study exhibits a noticeable gender skew, with approximately two-thirds of the participants being female.

Response: Thank you for pointing out this issue. In order to present the data as fully as possible, we did not screen the female data to balance the sex ratio. We added the limitation of the gender skew.

“Finally, the composition of the sample in our study has a gender skew, with approximately two-thirds of the participants being women. Caution is advised when inferring the results of NCT.”

  1. To my understanding, CPTSD is fundamentally characterized by its protracted duration. However, the absence of any reference to this temporal dimension in the study is noteworthy. The utilization of the Life Events Checklist for DSM-5 (LEC-5) in the research is acknowledged; however, its intended function, as delineated in its manual, is primarily the collection of data concerning potentially traumatic experiences. It does not proffer a formal scoring protocol or interpretation thereof. Hence, it remains unclear how the researchers identified and differentiated cases of CPTSD without factoring in the prolonged or repetitive nature of the traumatic experiences.

Response: Thank you for pointing out this issue. Unlike the fifth edition of the Diagnostic and Statistical Manual of Mental Disorders (DSM-5), the type of trauma is not a prerequisite for being diagnosed with CPTSD or PTSD in diagnostic criteria of ICD-11 (Daniunaite et al., 2021). As you said, previous studies found that CPTSD is more likely associated with prolonged and repeated trauma types, including those commonly happened in childhood such as physical abuse or sexual trauma (Brewin et al., 2017; Cloitre, 2020; Chiu et al., 2023). A study further suggests that adverse childhood experiences remained uniquely predictive of CPTSD symptom after controlling for lifelong traumatic stress. Some studies have also shown that some single life-changing accidents can cause CPTSD, because these trauma types may lead to violations of the integrity of the body, in turn, victims are unable to reexperience positive identities (Hyland et al., 2017; 2023).

Since this study focused on the impact of childhood trauma on CPTSD, some types of trauma that meets the criteria of DSM A criteria (such as disaster) may also lead to CPTSD, so we did not exclude these types. We added description of the impact of trauma types on CPTSD in the introduction.

“Unlike the fifth edition of the Diagnostic and Statistical Manual of Mental Disor-ders (DSM-5), the type of trauma is not a prerequisite for being diagnosed with CPTSD or PTSD in diagnostic criteria of ICD-11 [4]. Nevertheless, previous studies found that CPTSD is more likely associated with prolonged and repeated trauma types, including those commonly happened in childhood such as physical abuse or sexual trauma [5-7]. A study further suggests that adverse childhood experiences remained uniquely pre-dictive of CPTSD symptom after controlling for lifelong traumatic stress. Some studies have also shown that some single life-changing accidents can cause CPTSD, because these trauma types may lead to violations of the integrity of the body, in turn, victims are unable to reexperience positive identities [3,8].”

  1. According to the ICD-11, complex PTSD typically results from exposure to traumatic events of an extremely threatening nature, often characterized by their prolonged or repetitive nature and the impossibility of escape. I would appreciate more information on how the researchers established the presence of CPTSD in their sample, especially considering that the most frequent incidents reported were natural disasters, which generally have a lower probability of resulting in PTSD; transportation accidents, which are typically single events unless prolonged hospitalization or when injury or death of next of kin occurred; and physical assaults, which in many cases are isolated events.

Response: Thank you for pointing out this issue. Some studies have shown that some single life-changing accidents can cause CPTSD, because these trauma types may lead to violations of the integrity of the body, in turn, victims are unable to reexperience positive identities (Hyland et al., 2017; 2023).

We added the prevalence rates of probable CPTSD and probable PTSD based on the ITQ in results. Among 1416 participants with childhood trauma, the prevalence rates of probable CPTSD and probable PTSD were 9.96% (n = 141) and 5.86% (n = 83), respectively. We also reported prevalence of each CPTSD symptom, based on the cut-off of ITQ (scores ≥2 indicate the presence of a symptom).

“Among 1416 participants with childhood trauma, the prevalence rates of probable CPTSD and probable PTSD were 9.96% (n = 141) and 5.86% (n = 83), respectively. In addition, 9.60% (n = 136) of participants suffered from DSO symptoms.

 Table 2 shows the mean scores, standard deviations, and prevalence of the samples for each symptom of the ITQ. Long-term upset (AD1) was the most reported symptom, whereas feeling distant or cut off from others (DR1) were the least reported symptom.”

Table 2. Mean scores, standard deviations and prevalence of CPTSD symptoms

Symptom

M

SD

Percentage

RE1

1.00

1.10

28.46%

RE2

1.14

1.19

36.51%

AV1

1.48

1.27

44.07%

AV2

1.41

1.31

41.81%

TH1

1.02

1.24

30.16%

TH2

1.05

1.21

30.37%

AD1

1.60

1.06

47.46%

AD2

1.25

1.22

37.50%

NSC1

1.20

1.25

35.88%

NSC2

0.97

1.21

28.46%

DR1

0.98

1.15

27.40%

DR2

1.42

1.30

36.51%

Notes: RE1: nightmares; RE2: flashbacks; AV1: internal avoidance; AV2: external avoidance; TH1: hypervigilance; TH2: exaggerated startle response; AD1: long-term upset; AD2: emotional numbing; NSC1: feelings of failure; NSC2: feelings of worthlessness; DR1: feeling distant or cut off from others; DR2: difficulties feeling close to others

  1. Notwithstanding these reservations, I find this article to be intellectually stimulating, with the statistical methodologies employed therein both enlightening and innovative. Consequently, I anticipate the authors' forthcoming response addressing the concerns articulated above.

Response: Thank you for your careful reading and encouragement. We appreciate your comments and have revised the manuscript accordingly.

  1. Furthermore, I posit that the findings pertaining to gender-specific manifestations of CPTSD symptoms, notably the prevalence of nightmares and feelings of failure among men and the prevalence of hypervigilance, long-term distress, and exaggerated startle responses among women, carry immediate clinical relevance. This understanding could empower clinicians to deploy targeted interventions aimed at alleviating these symptoms, potentially enhancing the effectiveness of CPTSD treatments.

Response: Thank you for your careful reading and encouragement. We appreciate your comments and have revised the manuscript accordingly.

  1. Overall, I appreciate the article's contribution to the field and eagerly await the authors' response to these comments.

Response: Thank you for your careful reading and encouragement. We appreciate your comments and have revised the manuscript accordingly.

Thank you again for your time and effort, which helped immensely in improving this paper!

Reviewer 2 Report

Thank you very much for giving me the opportunity to review this manuscript. The idea of your article is interesting, my recommendations are the following:

Abstract

it would be recommended to explain how many men and women there are (differentiated), since the work is about differences by sex.

The exposure should be reviewed to be the first evidence of trauma and sex differences. It will be in China. Not on an international level.

 Introduction

It would be interesting to talk about explicitly starting from the WHO classification, since APA does not include it.

It would be interesting to explain that differences by sex are multifactorial (we cannot speak of biological chance, but rather of multiple influences, especially social influences such as sexism or gender).

Methods

It would be important to present all the inclusion/exclusion criteria in detail (line 100)

It would be important to specifically state what alpha was obtained in the adaptation to China and it is not said (line 109)

Results

Graphics of Figure 1 could be focused and sharpened

Graph 3.4. it is not completely understood

Discussion

It would be highly advisable to begin the discussion by exposing the main results in line with other studies, not to begin with the conclusion endorsing the study.

Conclusion

It would be advisable to develop conclusions

References

There is no reference to the current year

Author Response

Dear Reviewer 2,

Thank you very much for giving us the opportunity to revise our manuscript entitled “Sex differences in complex posttraumatic stress disorder network among Chinese young adults” (Manuscript Number: behavsci-2613111). We greatly appreciate your comments and suggestions. We have revised the manuscript and would like to resubmit it for publication in the Behavioral Sciences. We hope that the revised manuscript addresses most of the concerns that you raised and incorporates most of your suggestions. We hope to receive a favorable response from you. Please find our point-by-point responses attached.

Responses to Reviewer 2

Thank you very much for giving me the opportunity to review this manuscript. The idea of your article is interesting, my recommendations are the following:

Response: Thank you for your careful reading and encouragement. We appreciate your comments and have revised the manuscript accordingly.

Abstract: it would be recommended to explain how many men and women there are (differentiated), since the work is about differences by sex.

Response: Thank you for pointing out this issue. We added this information in abstract. In order to present the data as fully as possible, we did not screen the female data to balance the sex ratio. We also added the limitation of the gender skew.

“The current study utilized a representative sample of college students in Beijing and included 1416 participants (509 men, 907 women) who had childhood trauma experience.”

“Finally, the composition of the sample in our study has a gender skew, with approximately two-thirds of the participants being women. Caution is advised when inferring the results of NCT.”

The exposure should be reviewed to be the first evidence of trauma and sex differences. It will be in China. Not on an international level.

Response: Thank you for your valuable advice. We revised this expression according to your suggestion.

“The current study provides the first evidence of sex differences in the CPTSD symptoms network among Chinese young adults having childhood trauma.”

Introduction

It would be interesting to talk about explicitly starting from the WHO classification, since APA does not include it.

Response: Thank you for your valuable advice. According to your suggestion, we revised this paragraph. Since the purpose of this article is to introduce CPTSD, we started with the first proposing of CPTSD.

“Complex posttraumatic stress disorder (CPTSD) was first proposed by Her-man-Lewis in the study of women victims having experiencing chronic domestic and sexual violence [1]. Under discussion over the years, the World Health Organization introduced CPTSD as an independent diagnosis in the 11th Revision of International Classification of Diseases (ICD-11)[2]. In the diagnostic criteria of trauma-related mental disorders, the ICD-11 introduced CPTSD as a sibling disorder to posttraumatic stress disorder (PTSD).”

It would be interesting to explain that differences by sex are multifactorial (we cannot speak of biological chance, but rather of multiple influences, especially social influences such as sexism or gender).

Response: Thank you for your valuable advice. We revised this expression according to your suggestion.

“Sex differences in trauma-related disorders are multifactorial, differences in the type of trauma that men and women are likely to be exposed to and their reactivity after trauma are important social psychological factors [22,23]”

Methods

It would be important to present all the inclusion/exclusion criteria in detail (line 100)

Response: Thank you for your valuable advice. We added detail information.

“A total of 2048 participants from 29 universities completed the survey. Among them 221 participants were excluded due to careless answering (e.g., answering the same answer to each item or failure to pass the attention check items). After that, 1827 (89.2%) valid data were furth er screened: 411 participants who didn’t report childhood trauma history was further excluded according to the Life Events Checklist for DSM-5. Finally, 1416 met the inclusion criteria (i.e., having childhood trauma and being aged 18-28 years).”

It would be important to specifically state what alpha was obtained in the adaptation to China and it is not said (line 109)

Response: Thank you for your valuable advice. We added this information.

“The Chinese version of the ITQ was translated by Ho and colleagues and has been demonstrated to have good reliability and validity (Cronbach’s α of the six PTSD items = 0.89; Cronbach’s α of the six DSO items = 0.90)”

Ho, G. W., Karatzias, T., Cloitre, M., Chan, A. C., Bressington, D., Chien, W. T., ... & Shevlin, M. (2019). Translation and validation of the Chinese ICD-11 international trauma questionnaire (ITQ) for the assessment of posttraumatic stress disorder (PTSD) and complex PTSD (CPTSD). European Journal of Psychotraumatology, 10(1), 1608718.

Results

Graphics of Figure 1 could be focused and sharpened

Response: Thank you for your valuable advice. We've centered this figure and provide a clearer version.

Graph 3.4. it is not completely understood

Response: Thank you for pointing out this issue. We explain more about Figure 4 in the results.

“The Bayesian network was used to predict the probable direction of edges in men and women networks. Upstream variables are seen as the causes of downstream variables in Bayesian network, so upstream symptoms have a greater likelihood of activating the whole network.”

Discussion

It would be highly advisable to begin the discussion by exposing the main results in line with other studies, not to begin with the conclusion endorsing the study.

Response: Thank you for your valuable advice. We changed the order of discussion based on your advice.

“Both men and women have higher strength centrality in negative self-concept symptoms of DSO, feeling of failure was among the more central nodes both in men and women CPTSD network. Previous study also found negative self-concept of DSO was the central symptom in CPTSD network [15,16], which have previously been considered central to the diagnosis of CPTSD [31]. Negative self-evaluations, such as self-deprecation and self-loathing, were observed to be associated with a higher risk in individuals with childhood trauma experience [3]. These people scored lower on emo-tional self-esteem as well [47,48].”

Conclusion

It would be advisable to develop conclusions

Response: Thank you for your valuable advice. We added the conclusion section.

“5 Conclusion

The current study found that the sex differences in the CPTSD networks were mainly manifested in the strength centrality and bridge strength of symptoms. Nightmares and feelings of failure had the highest strength centrality for men, while hypervigilance and feelings of failure had the highest strength centrality for women. Long-term upset and nightmares had the highest bridge centrality for men, while long-term upset and exaggerated startle response had the highest bridge centrality for women. According to these findings, attention should also be paid to different CPTSD symptoms according to sex in clinical work.”

References

There is no reference to the current year

Response: Thank you for pointing out this issue. We added some references in this year.

“Chiu, H.T.S.; Alberici, A.; Claxton, J.; Meiser-Stedman, R. The prevalence, latent structure and psychosocial and cognitive correlates of complex post-traumatic stress disorder in an adolescent community sample. J. Affect. Disord. 2023, 340, 482-489, doi:10.1016/j.jad.2023.08.033.

Hyland, P.; Shevlin, M.; Brewin, C.R. The Memory and Identity Theory of ICD-11 Complex Posttraumatic Stress Dis-order. Psychol. Rev. 2023, 130, 1044-1065, doi:10.1037/rev0000418.”

Thank you again for your time and effort, which helped immensely in improving this paper!
